# The Generalized Gamma Distribution as a Useful RND under Heston's Stochastic Volatility Model

Benzion Boukai 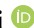

Department of Mathematical Sciences, Indiana University—Purdue University Indianapolis (IUPUI), Indianapolis, IN 46202, USA; bboukai@iupui.edu; Tel.: +1-317-274-6926

**Abstract:** We present the Generalized Gamma (GG) distribution as a possible risk neutral distribution (RND) for modeling European options prices under Heston's stochastic volatility (SV) model. We demonstrate that under a particular reparametrization, this distribution, which is a member of the scale-parameter family of distributions with the mean being the forward spot price, satisfies Heston's solution and hence could be used for the direct risk-neutral valuation of the option price under Heston's SV model. Indeed, this distribution is especially useful in situations in which the spot's price follows a negatively skewed distribution for which Black–Scholes-based (i.e., the log-normal distribution) modeling is largely inapt. We illustrate the applicability of the GG distribution as an RND by modeling market option data on three large market-index exchange-traded funds (ETF), namely the SPY, IWM and QQQ as well as on the TLT (an ETF that tracks an index of long-term US Treasury bonds). As of the writing of this paper (August 2021), the option chain of each of the three market-index ETFs shows a pronounced skew of their volatility 'smile', which indicates a likely distortion in the Black–Scholes modeling of such option data. Reflective of entirely different market expectations, this distortion in the volatility 'smile' appears not to exist in the TLT option data. We provide a thorough modeling of the option data we have on each ETF (with the 15 October 2021 expiration) based on the GG distribution and compare it to the option pricing and RND modeling obtained directly from a well-calibrated Heston's SV model (both theoretically and also empirically, using Monte Carlo simulations of the spot's price). All three market-index ETFs exhibited negatively skewed distributions, which are well-matched with those derived under the GG distribution as RND. The inadequacy of the Black–Scholes modeling in such instances, which involves negatively skewed distribution, is further illustrated by its impact on the hedging factor, delta, and the immediate implications to the retail trader. Similarly, the closely related Inverse Generalized Gamma distribution (IGG) is also proposed as a possible RND for Heston's SV model in situations involving positively skewed distribution. In all, utilizing the Generalized Gamma distributions as possible RNDs for direct option valuations under the Heston's SV is seen as particularly useful to the retail traders who do not have the numerical tools or the know-how to fine-calibrate this SV model.

**Keywords:** heston model; option pricing; risk-neutral valuation; calibration; volatility skew; negatively skewed distribution; market data (SPY; QQQ; IWM; TLT)

**JEL Classification:** G10; G13

## 1. Introduction

One of the most widely celebrated option pricing models for equities (and beyond) is that of Black and Scholes (1973) and of Merton (1973) (abbreviated here as the BSM model). Their option pricing model was derived under some simple assumptions concerning the distribution of the asset's returns, coupled with presumptive continuous hedging, self-financing, zero dividend, risk-free interest rate, $r$, and no cost of carry or transactions fees. In its standard form, the BSM model assumes that the spot's price process $\mathcal{S} = \{S_t, t \geq 0\}$ evolves with a constant volatility of the spot's returns, $\sigma$, as a geometric Brownian motion

(under a risk-neutral probability measure $\mathbb{Q}$, say), leading to an exact solution for the price, $C(\cdot)$, of an European call option. Specifically, given the current spot price $S_\tau = S$ and the risk-free interest rate $r$, the price of the corresponding call option with price-strike $K$ and duration $T$,

$$C_S(K) = S\,\Phi(d_1) - K\,e^{-rt}\,\Phi(d_2), \tag{1}$$

where $t = T - \tau$ is the lremaining time to expiry. Here, we use the conventional notation to denote by $\Phi(\cdot)$ and $\phi(\cdot)$ the standard normal cumulative distribution function (*cdf*) and density function (*pdf*), respectively, and where

$$d_1 := \frac{-\log(\frac{K}{S}) + (r + \frac{\sigma^2}{2})t}{\sigma\sqrt{t}} \quad \text{and} \quad d_2 := d_1 - \sigma\sqrt{t}. \tag{2}$$

Despite its wide acceptability in the retail trading world[1], this model hinges on several incorrect assumptions and hence suffers from some notable deficiencies; see for example Black (1989) who pointed out 'the holes in Black–Scholes'. Chief among these deficiencies is the fact that the volatility of a spot's returns (i.e., $\sigma$) appears not to be constant over the 'life' of the option but rather varying at random.

The efforts to incorporate a non-constant volatility term in the option valuation (e.g., Wiggins 1987 or Stein and Stein 1991) has culminated with stochastic volatility (SV) model introduced by Heston (1993) (see (A1)). This SV model incorporates, aside from the dynamics of the spot's price process $\mathcal{S}$, also the dynamics of a corresponding, though unobservable (hence untradeable), volatility process $\mathcal{V} = \{V_t, t \geq 0\}$. Instructed by the *form* of the exact BSM solution in (1), Heston (1993) obtained that the solution to the system of PDE he obtained from the stochastic volatility model he constructed is given by

$$C_S(K) = S\,P_1 - K\,e^{-rt}\,P_2, \tag{3}$$

where $P_j\ j = 1, 2$ are two related (under $\mathbb{Q}$) conditional probabilities that the option will expire in-the-money, conditional on the given current stock price $S_\tau = S$ and the current volatility, $V_\tau = V_0$. However, unlike the explicit BSM solution in (1) which is given in terms of the normal (or log-normal) distribution, Heston (1993) provided (semi) closed-form solutions to these two probabilities, $P_1$ and $P_2$, which were both given in terms of their characteristic functions (*c.f.*); see (A2). These characteristic functions depend on some parameters of the SV model, $\vartheta = (\kappa, \theta, \eta, \rho)$, and they may be evaluated numerically for any choice of the parameters $\vartheta$, in addition to the given $S$, $V_0$ and $r$ (for more details, see Appendix A). The components of $\vartheta$ have particular meaning in the context of Heston's SV model: $\rho$ is the correlation between the random components of the spot's price and volatility processes, $\theta$ is the long-run average volatility, $\kappa$ is the mean-reversion speed for the volatility dynamics and $\eta^2$ is the variance of the volatility $V$. It should be noted that different choices of $\vartheta$ will lead to different *values* $C_S(K)$ in (3) and hence, the value $\vartheta = (\kappa, \theta, \eta, \rho)$ must be appropriately *'calibrated'* first for $C_S(K)$ to actually match the option market data. However, this calibration process typically involves substantial numerical challenges (largely resulting from numerical issues involved in the required multi-dimensional optimization, see for example Bin 2007, or Section 2.1 in Romo and Ortiz-Gracia 2021). These challenges are an obvious hindrance to the retail option traders who do not have the numerical tools or the know-how to finely calibrate the Heston (1993) SV model, as needed in the evaluation of $C_S(K)$ in (3).

On the other hand, as was established by Cox and Ross (1976), the risk-neutral option valuation (under $\mathbb{Q}$) provides that for $T > \tau$ (with $t = T - \tau$), the option price $C_S(K)$ must also satisfy

$$
\begin{aligned}
C_S(K) &= e^{-rt} \int_K^\infty (S_T - K)\, q(S_T) dS_T, \\
&= e^{-rt} \int_K^\infty S_T q(S_T) dS_T - K e^{-rt} \int_K^\infty q(S_T) dS_T \\
&\equiv e^{-rt} \int_K^\infty S_T q(S_T) dS_T - K e^{-rt} \cdot (1 - Q(K)),
\end{aligned}
\tag{4}
$$

where $q(\cdot)$ is the density of some risk-neutral distribution (RND) $Q(\cdot)$, under the probability $\mathbb{Q}$, reflective of the conditional distribution of the spot price $S_T$ at time $T$, given the spot price, $S_\tau$ at time $\tau < T$, whose expected value is the future value of the spot's price. Namely, the RND $q(\cdot)$ must also satisfy,

$$
\mathbb{E}(S_T | S_\tau = S) = \int S_T \cdot q(S_T) dS_T = S \cdot e^{rt}.
\tag{5}
$$

This risk-neutral density (or distribution) links together for the option valuation (under $\mathbb{Q}$) the distribution of the spot's price $S_T$ and the stochastic dynamics governing the underlying model. As was mentioned earlier, in the case of the BSM model in (2), the RND is unique and is given by the log-normal distribution. However, since Heston's SV model involves the dynamics of two stochastic processes, one of which (the volatility process, $\mathcal{V}$) is untradeable and hence not directly observable, there are innumerable many possible choices of RNDs, $q(\cdot)$, that would satisfy (4) and (5), and hence, the general solutions of Heston's $P_1$ an $P_2$ in (3) are as given by means of their characteristic functions (A2), per each possible choice of the structural parameter $\vartheta = (\kappa, \theta, \eta, \rho)$.

### 1.1. Heston's RND as a Class of Scale-Parameter Distributions

In the literature, one can find numerous papers dealing with the 'extraction', 'recovery', 'estimation' or 'approximation', in parametric or non-parametric frameworks, or even within an entropy-based pricing framework (e.g., Yu 2020) of the RND, $q(\cdot)$ from the available (market) option prices. Some comprehensive literature reviews of the subject can be found in Jackwerth (2004); Figlewski (2010); Girth and Krätschmer (2012) and Figlewski (2018). In particular, within the parametric approach, one attempts to estimate by various standard means (maximum likelihood, method of moments, least squares, etc.) the parameters of some *assumed* distribution so as to approximate available option data or implied volatility (cf. Jackwerth and Rubinstein 1996). This type of *assumed* multi-parameter distributions includes some mixtures of log-normal distribution Mizrach (2010); Girth and Krätschmer (2012), generalized gamma Girth and Krätschmer (2012), generalized extreme value Figlewski (2010), the gamma and the Weibull distributions (Savickas 2005), among others. While empirical considerations have often led to suggesting these particular parametric distributions as possible *pdf* in (4), the motivation for these considerations did not include any direct link to the governing pricing model and its dynamics especially in the case of Heston's SV model in (A1). As was mentioned earlier, in the case in the BSM model, linking directly the log-normal distribution and the price dynamics reflected by the geometric Brownian motion led to the BSM formula in (1).

In the case of the Heston (1993) SV model, this direct link between the governing price and volatility dynamics of $(S, \mathcal{V})$ in (A1) and the class of distributions that could serve as RNDs for it has been established in Boukai (2021).

Let $\Delta(K)$ be the so-called *delta* function (or hedging fraction) in the option valuation, as defined by

$$
\Delta(K) = \frac{\partial C_S(K)}{\partial S}.
\tag{6}
$$

It is well-known that for Heston's solution for the call option price $C_S(K)$, $P_1 \equiv \Delta(K)$ in (3); see for example Bakshi et al. (1997) or Boukai (2021). Hence, accounting also for (4), it

follows that under the SV model (A1), Heston's solution for the option price in (3) can be written in an equivalent form as

$$C_S(K) \equiv S \cdot \Delta(K) - K e^{-rt} \cdot (1 - Q(K)), \tag{7}$$

It was shown in Boukai (2021) that the general class of scale-parameter distributions that satisfies Assumption 1 below, with the mean being the forward spot's price, $\mu := S \cdot e^{rt}$ would admit the presentation in (7) and hence would satisfy Heston's solution for the option price in (3). In fact, it was also shown that the actual RNDs (see (A4)) that may be calculated directly from Heston's characteristic functions (A2), of to $P_1$ and $P_2$ (see Appendix A) are members of this class of distributions as well. Accordingly, the main results of Boukai (2021) (as are summarized in Theorem 2 there) establish the direct link through the solution of Heston (1993) in (3) (or (7)) between this class of RNDs and the assumed stochastic volatility model governing the spot price and volatility dynamics.

To fix ideas, we set $\mu = S \cdot e^{rt}$ to denote the forward spot's price, and correspondingly, we denote by $Q_\mu(\cdot)$ the RND with a corresponding *pdf* $q_\mu(\cdot)$ as in (4) and (5).

**Assumption 1.** *It is assumed that $\mu$ is a scale parameter of $Q_\mu(\cdot)$ so that for any $x > 0$, $Q_\mu(x) \equiv Q_1(x/\mu)$ and $q_\mu(x) \equiv q_1(x/\mu)/\mu$ for some cdf $Q_1(\cdot)$ with a pdf $q_1(\cdot)$ satisfying $\int_0^\infty x q_1(x)dx = 1$ and $\int_0^\infty x^2 q_1(x)dx = 1 + v^2$. Here, $v^2$ is some exogenous parameter (to be specified later).*

Note that if $Q_\mu(\cdot)$ satisfies Assumption 1, then, for any $k > 0$,

$$\begin{aligned} c_\mu(k) &:= \int_k^\infty (x - k) q_\mu(x)dx = \int_k^\infty (1 - Q_\mu(x))dx \\ &= \int_k^\infty (1 - Q_1(x/\mu))dx \equiv \mu\, c_1(k/\mu). \end{aligned} \tag{8}$$

Hence, $c_\mu(k)$ is a homogeneous function[2] of degree one in both $\mu$ and $k$, so that for $k' = a k$ and $\mu' = a \mu$ with $a > 0$, we have $c_{\mu'}(k') \equiv a c_\mu(k)$. This property and (8) are the key elements in the proof of Theorem 1 of Boukai (2021), which establishes that for this class of scale-parameter distributions satisfying Assumption 1, the delta function (6) may in fact be written as

$$\Delta(K) \equiv \Delta_\mu(K) := \frac{1}{\mu} \int_K^\infty x q_\mu(x)dx \equiv \Delta_1(K/\mu), \tag{9}$$

where $\Delta_1(a) := \int_a^\infty u q_1(u)du$ for any $a > 0$. Accordingly, for any member of this scale parameter (in $\mu = S \cdot e^{rt}$) class of distributions defined by $Q_1(\cdot)$ as in Assumption 1, the option price $C_S(K)$ in (7) may equivalently be written as

$$C_S(K) \equiv S \cdot \Delta_1(K/\mu) - K e^{-rt} \cdot (1 - Q_1(K/\mu)), \tag{10}$$

and thus, it could be used for the direct risk-neutral valuation, as RND, of the option price under Heston's SV model. The expression in (10) is our 'working' formula for the direct calculations of the option price $C_S(K)$ in the case of scale-parameter distribution defined by $Q_1(\cdot)$. This result was illustrated in great detail by Boukai (2021) for several well-known parametric (scale) distributions which under a particular parametrization satisfy Assumption 1. These distributions include one-parameter versions of the log-Normal (i.e., the BSM model), Inverse-Gaussian, Gamma, Inverse-Gamma, Weibull and the Inverse-Weibull distributions, all which provide explicit RNDs for Heston's pricing model in various market circumstances (e.g., negatively skewed RND to match SPX option data (i.e., Bakshi et al. 1997), or ODAX option data (i.e., Mrázek and Pospíšil 2017), or positively skewed RND to match AMD (say), option data).

**Remark 1.** *In the case in which the risk-neutral valuation of the option includes a dividend with a rate $\ell$, then $\mathbb{E}(S_T | S) = S e^{(r-\ell)t}$ in (5), in which case, by applying $\mu = S e^{(r-\ell)t}$ to (7), we obtain from (8) and (10),*

$$C_S(K) = e^{-rt} c_\mu(K) = S e^{-\ell t} \Delta_1(K/\mu) - K e^{-rt} (1 - Q_1(K/\mu)).$$

*1.2. An Overview*

In this paper, we focus attention on a two-parameter version of the Generalized Gamma (GG) distribution as is especially parametrized to satisfy Assumption 1 and hence, to serve as an RND under Heston's SV option valuation model. The particular version of this distribution we consider here is characterized by two shape parameters $\alpha$ and $\xi$ say, and it is general enough to admit either positively skewed distributions ($\xi < 0$) or negatively skewed distributions ($\xi > 0$). Aside from this noted 'elasticity' to match well the varying characteristics of different spot's RNDs under the SV model (A1) implied from different market scenarios (i.e., with different $\vartheta = (\kappa, \theta, \eta, \rho)$ in (A1)), this distribution is especially useful in modeling option prices in situations that exhibit put-over-call skew and and hence admit negatively skewed distribution of the spot's price, indeed with $\xi > 0$.

In Section 2, we present the GG distribution and its required reparametrization as a RND for the SV model (3). Though not of immediate interest, we also present in Section 2.2 the case with $\xi < 0$ (the so-called Inverse Generalized Gamma (IGG) distribution) as a possible RND under Heston's SV model that could be useful in modeling positively skewed (implied) distributions.

In Section 3, we apply the GG distribution (with $\xi > 0$) as RND to modeling current[3] market option data on three large market-index ETFs, namely the SPY, IWM and QQQ as well as on the TLT (a large ETF that tracks an index of long-term US Treasury bonds). The current option chain of each of the three market ETFs exhibits a pronounced skew of their volatility 'smile', which indicates a likely distortion in the Black–Scholes modeling of such option data. Reflective of entirely different market expectations, this distortion appears not to exist in the TLT option data (see Figure 1 below). We provide a thorough modeling of the available option data we have on each ETF (with the 15 October 2021 with 63 days to expiration) based on the GG distribution (with $\xi > 0$) and compare it to the option pricing and RND modeling obtained directly from a well-calibrated Heston (1993) SV model (both theoretically and empirically, using Monte Carlo simulations of the spot's price). All three market-index ETFs exhibit negatively skewed distributions, which are well-matched with those derived under the GG distribution as RND. The inadequacy of the classical Black–Scholes modeling in such instances which involve negatively skewed implied distribution is further illustrated by its impact on the hedging factor, delta, and the immediate implications to the retail trader. In contrast, for the TLT ETF, which exhibits no such distortion to the volatility 'smile', the three pricing models (i.e., Heston's, Black–Scholes and Generalized Gamma) appear to yield very similar results. Technical notes are provided in Section 3.2, and some details on Heston's SV model and related cf. that are used in the calculation of (3) are provided in Appendix A.

## 2. The Generalized Gamma Distribution as an RND for Heston's SV Model

Introduced by Stacy (1962), the Generalized Gamma (GG) distribution is demonstrably highly versatile, with a vast number applications, from survival analysis to meteorology and beyond. It includes among many others the Weibull distribution ($\alpha = 1$), the Gamma distribution ($\xi = 1$), and also the log-normal distribution as a limiting case, ($\alpha \to \infty$). In this section, we show that this GG distribution along with its counterpart, the so-called Inverse Generalized Gamma distribution (IGG), both satisfy under a particular reparametrization, Assumption 1, and hence could serve as RND (for direct option valuation using (10)) under the Heston (1993) SV model for option valuation. Though similar, we will present these two cases of the Generalized Gamma distribution separately as they do present different profiles of skewness and kurtosis. We will however focus our attention on the GG distribution

(with $\xi > 0$), as we will use it for option pricing modeling in a situation that involved negatively skewed (implied) risk-neutral distributions.

We begin with some standard notations. We write $Y \sim \mathcal{G}(\alpha, \lambda)$ to indicate that the random variable $Y$ has the gamma distribution with a scale parameter $\lambda > 0$ and a shape parameter $\alpha > 0$ (so that its mean is $\mathbb{E}(Y) = \alpha/\lambda$). We write $g(\cdot; \alpha, \lambda)$ and $G(\cdot; \alpha, \lambda)$ for the corresponding *pdf* and *cdf* of $Y$, respectively,

$$g(y; \alpha, \lambda) \equiv \frac{\lambda^\alpha y^{\alpha-1} e^{-\lambda y}}{\Gamma(\alpha)} \qquad \text{and} \qquad G(y; \alpha, \lambda) \equiv \frac{\Gamma(y\lambda; \alpha)}{\Gamma(\alpha)}, \tag{11}$$

where $\Gamma(\alpha) := \int_0^\infty x^{\alpha-1} e^{-x} dx$ denotes the gamma function whose incomplete version is $\Gamma(s; \alpha) := \int_0^s x^{\alpha-1} e^{-x} dx$, is defined for any $s > 0$.

### 2.1. The GG Distribution

The Generalized Gamma (GG) distribution is typically characterized by three parameters: a scale parameter, $\lambda > 0$, and two shape parameters, $\alpha > 0$ and $\xi > 0$, and it is defined as follows. We say that $W \sim \mathcal{GG}(\lambda, \xi, \alpha)$, if

$$Y \equiv \left(\frac{W}{\lambda}\right)^\xi \sim \mathcal{G}(\alpha, 1). \tag{12}$$

In light of relation (12), the *cdf* and *pdf* of $W \sim \mathcal{GG}(\lambda, \xi, \alpha)$ are readily available in terms of the Gamma distribution in (11). More specifically, for any $w > 0$,

$$F_W(w) := Pr(W \leq w) = G\left(\left(\frac{w}{\lambda}\right)^\xi; \alpha, 1\right),$$

and

$$f_W(w) = \frac{\xi}{\lambda}\left(\frac{w}{\lambda}\right)^{\xi-1} \cdot g\left(\left(\frac{w}{\lambda}\right)^\xi; \alpha, 1\right).$$

In addition, the $j$th, $j = 0, 1, 2, \ldots$, moment of this distribution (see Stacy and Mihram 1965), whenever it exists, (i.e., whenever $\alpha + j/\xi > 0$) is given by

$$\mathbb{E}(W^j) = \lambda^j \frac{\Gamma(\alpha + j/\xi)}{\Gamma(\alpha)} := \lambda^j \cdot h_j(\xi), \quad \text{with} \quad \alpha + j/\xi > 0, \quad \text{and} \quad \alpha > 0. \tag{13}$$

Now, suppose that for a given $\alpha > 0$, a random variable $U$ has the 'standardized' version of the GG distribution, with mean $\mathbb{E}(U) = 1$ and a variance $Var(U) = \nu^2$, for some $\nu > 0$ (in fact, we will later take $\nu = \sigma\sqrt{t}$ for some $\sigma > 0$). Utilizing $h_j(\xi)$ as defined in (13), we let for a given $\alpha > 0$ and $\nu > 0$, $\xi^* \equiv \xi(\nu)$ be the (unique) solution of the equation

$$\frac{h_2(\xi)}{h_1^2(\xi)} = 1 + \nu^2, \tag{14}$$

in which case, $h_j^* \equiv h_j(\xi^*)$, $j = 1, 2$, $\lambda^* \equiv 1/h_1^*$ and $U \sim \mathcal{GG}(\lambda^*, \xi^*, \alpha)$. Accordingly, the *cdf* of $U$ is given by

$$Q_1(u) := Pr(U \leq u) = G\left(\left(\frac{u}{\lambda^*}\right)^{\xi^*}; \alpha, 1\right), \tag{15}$$

for any $u > 0$, and its *pdf* is given by,

$$q_1(u) := \frac{\xi^*}{\lambda^*}\left(\frac{u}{\lambda^*}\right)^{\xi^*-1} \cdot g\left(\left(\frac{u}{\lambda^*}\right)^{\xi^*}; \alpha, 1\right), \quad u > 0. \tag{16}$$

It follows immediately from (15) that if $X \equiv \mu \cdot U$ for some $\mu > 0$, then the *pdf*, $q_\mu(\cdot)$ of $X$ is the 'scaled' version of $q_1(\cdot)$ above. For this RND, the values of $\Delta_1(s)$ in (9) can be calculated using (16) for any $a > 0$ as,

$$\Delta_1(a) = \int_a^\infty u q_1(u) du = 1 - G((a/\lambda^*)^{\xi^*}; \alpha + 1/\xi^*, 1), \tag{17}$$

which, when combined in (10) with the expression of $Q_1(\cdot)$ as given in (15) above, provides the values of

$$
\begin{aligned}
c_\mu(k) &= \mu \times \left[ \Delta_1(k/\mu) - \frac{k}{\mu} \times (1 - Q_1(k/\mu)) \right], \\
&= \mu \times \left[ 1 - G((k/\mu\lambda^*)^{\xi^*}; \alpha + 1/\xi^*, 1) \right] - k \times \left[ 1 - G((k/\mu\lambda^*)^{\xi^*}; \alpha, 1) \right]
\end{aligned}
\tag{18}
$$

for any $\mu > 0$. Finally, to calculate under this Generalized Gamma RND the price of a call option at a strike $K$ when the current price of the spot is $S$, we will utilize (18) with $\mu \equiv S e^{rt}$ (being the forward price), $k \equiv K$ and with $\lambda^* \equiv 1/h_1(\xi^*)$ and $\xi^* \equiv \xi(\nu)$ as is determined by Equation (14) above with $\nu \equiv \sigma\sqrt{t}$ to obtain $C_S(K) = e^{-rt} c_\mu(K)$ as

$$C_S(K) = S \cdot [1 - G(d; \alpha + 1/\xi^*, 1)] - Ke^{-rt} \cdot [1 - G(d; \alpha, 1)], \tag{19}$$

where

$$d = \left( \frac{Ke^{-rt} h_1(\xi^*)}{S} \right)^{\xi^*}, \qquad \text{with} \qquad \xi^* \equiv \xi(\nu) \quad \text{from (14)}.$$

We point out that for given current spot's price, $S$, a strike price $K$, risk-free interest rate, $r$, and the remaining option's duration $t$, the option value $C_S(K)$ in (19) involves, through Equation (14) (with $\nu \equiv \sigma\sqrt{t}$), with only two parameters, namely $\alpha$ and $\sigma$. Their values can easily be "*calibrated*" from the available market option data. Indeed, in the Generalized Gamma case, this calibration task is computationally much simpler than the direct calibration of four parameters, $\vartheta = (\kappa, \theta, \eta, \rho)$, of Heston's pricing model, based on the characteristic functions (see (A2) in Appendix A), which also involves integration over the complex domain.

## 2.2. The IGG Distribution

For the sake of completeness, we also present the details of this variant to the Generalized Gamma distribution here as well. With some additional restrictions on $\xi$, one can similarly define the Inverse Generalized Gamma distribution (IGG). Namely, we say that $W \sim \mathcal{IGG}(\lambda, \xi, \alpha)$, if

$$Y \equiv \left( \frac{W}{\lambda} \right)^{-\xi} \sim \mathcal{G}(\alpha, 1). \tag{20}$$

The option pricing model under the Inverse Generalized Gamma distribution as RND for the Heston's SV for option valuation is constructed similarly to that of the GG in the previous section. By relation (20), if $W \sim \mathcal{IGG}(\lambda, \xi, \alpha)$, then its *cdf* is given, for $w > 0$,

$$F_W(w) := Pr(W \le w) = 1 - G\left( \left( \frac{w}{\lambda} \right)^{-\xi}; \alpha, 1 \right).$$

In this case, too, the 'standardized' IGG distribution of $U$ is constrained to have mean 1 and variance $\nu^2$, which requires a restriction on the parameter $\xi > 2/\alpha$. It follows that with such a restriction, $U \sim \mathcal{IGG}(\lambda^*, \xi^*, \alpha)$, but now, $\xi^* \equiv \xi(\nu)$ is the (unique) solution of the equation

$$\frac{\tilde{h}_2(\xi)}{\tilde{h}_1^2(\xi)} = 1 + \nu^2, \tag{21}$$

where, $\tilde{h}_j(\xi) \equiv h_j(-\xi) = \Gamma(\alpha - j/\xi)/\Gamma(\alpha)$, $j = 1, 2$, provided that $\alpha - j/\xi > 0$, in which case, $\tilde{h}_j^* \equiv \tilde{h}_j(\xi^*)$, $j = 1, 2$, $\lambda^* \equiv 1/\tilde{h}_1^*$. Accordingly, the *cdf* of $U$ is given by

$$Q_1(u) := Pr(U \le u) = 1 - G\left(\left(\frac{u}{\lambda^*}\right)^{\xi^*}; \alpha, 1\right), \tag{22}$$

for any $u > 0$, and in similarity to (17), its corresponding delta function is given by

$$\Delta_1(s) = G((s/\lambda^*)^{-\xi^*}; \alpha - 1/\xi^*, 1), \tag{23}$$

Again, by combining (22) and (23) in (10), we obtain that for any $\mu > 0$,

$$c_\mu(k) = \mu \times G((k/\mu\lambda^*)^{-\xi^*}; \alpha - 1/\xi^*, 1) - k \times G((k/\mu\lambda^*)^{-\xi^*}; \alpha, 1). \tag{24}$$

Accordingly, in order to calculate under this Inverse Generalized Gamma RND the price of a call option at a strike $K$ when the current price of the spot is $S$, we will utilize (24) with $\mu \equiv S\,e^{rt}$, $k \equiv K$ and with $\lambda^* \equiv 1/\tilde{h}_1(\xi^*)$ and $\xi^* \equiv \xi(\nu)$ as is determined by Equation (21) above with $\nu \equiv \sigma\sqrt{t}$ to obtain, $C_S(K) = e^{-rt}c_\mu(K)$ as,

$$C_S(K) = S \cdot G(d; \alpha - 1/\xi^*, 1) - Ke^{-rt} \cdot G(d; \alpha, 1), \tag{25}$$

where

$$d = \left(\frac{Ke^{-rt}\tilde{h}_1(\xi^*)}{S}\right)^{-\xi^*}, \qquad \text{with} \qquad \xi^* \equiv \xi(\nu) \quad \text{from (21).}$$

### 2.3. Skew and Kurtosis

As can be seen from the above construction of the RNDs, both the GG and IGG distributions depend on two shape parameters $(\alpha, \xi^*)$, or equivalently $(\alpha, \nu)$, where $\nu \equiv \sigma\sqrt{t}$, that affect their features, such as *kurtosis* and *skewness*, and hence their suitability as RND for various particular scenarios of the SV model (A1), as is determined by the structural model parameter $\vartheta = (\kappa, \theta, \eta, \rho)$ (more on this point in the next section). Unlike the standardized log-normal distribution which has a positive skew only, these two classes of distributions offer a range of RNDs with positive as well as negative skewness. This is a critical feature to have when modeling option prices for characteristically different spots such as an Index (SPX, say) as opposed to modeling option prices for a technology firm (such as AMD, say).

For a given $(\alpha, \xi^*)$, we denote these two measures as $\gamma_1(\xi^*)$ for *skew* and $\gamma_2(\xi^*)$ for the *kurtosis*. Then, with $h_j(\xi) := \Gamma(\alpha + j/\xi)$, we have in the GG case that with $\xi^* = \xi(\nu)$ which satisfies (14),

$$\gamma_1(\xi^*) = \frac{h_3(\xi^*) - 3\nu^2 - 1}{\nu^3}$$

and

$$\gamma_2(\xi^*) = \frac{h_4(\xi^*) - 4\nu^3\gamma_1(\xi^*) - 6\nu^2 - 1}{\nu^4}.$$

For the IGG case, these two measure are similar and are given by $\gamma_1(-\xi^*)$ and $\gamma_2(-\xi^*)$, provided that $\xi^* = \xi(\nu)$ as is determined by (21) satisfies that $\xi^* > 4/\alpha$.

### 3. Calibration, Validation and Examples

#### 3.1. Observing the Skew

In this section, we demonstrate the usefulness of the Generalized Gamma distribution to serve as an RND under Heston's Stochastic Volatility model in cases that exhibit a high put–call skew (i.e., OTM puts in the option series are far more expensive than equidistant OTM calls) and hence expressing a pronounced skew in the so-called "volatility smile" of the series. Cases in point are traded market indexes such as the S&P 500 (SPX), Russel 2000 (RUT) or Nasdaq 100 (NDX), which all are (along with their corresponding ETF surrogates,

SPY, IWM and QQQ) currently at (or near) their all time high levels[4]. Market expectations of an eminent 'correction' are often seen as the culprits that affected the implied volatility surface associated with the corresponding option series of the index (see for example Bakshi et al. 1997).

Figure 1 below displays the calculated implied volatility smile of the 15 October 2021 option series for these three ETFs, SPY, IWM and QQQ, as quoted on 13 August 2021, (EOD), each with 63 days to expiration (DTE). Several days later, on 18 August 2021, we obtained the corresponding quote for the TLT, but now with 57 DTE. For each ETF, the EOD option's market prices (for puts and calls) at the corresponding strikes were recorded along with the BSM-based calculated delta and implied volatility as provided by the brokerage firm.[5] As a reference, we also marked on these plots (in red) the current spot's (ETF) price *S* along with the ATM (BSM-based) calculated implied volatility (IV) for each ETF. As can be seen from these figures, the options of the three market index ETFs exhibit a highly pronounced skew in their volatility 'smile', whereas the option on the TLT ETF does not (likely only reflective of market's expectations of actions by the Federal Reserve).

However, since typically in the retail world, the calculated option's implied volatility (as well as other associated quantities, such as the option's delta) is calculated based on the Black–Scholes formula in (1), the noted distortion in the volatility smile (or surface) is nonetheless also indicative that the assumed underlying log-normal distribution of the Black–Scholes model (with its distinctive positive skew measure) is a poor choice to serve as RND in such instances involving a stochastic volatility structure as that of Heston (1993) (see (A1) below), particularly in those instances that admit a negatively skewed RND. To illustrate the extent of the "inaptness" of using the log-normal distribution as RND (the BSM formula in (1)) for the option valuation in such skewed cases, we have *calibrated* for each of these four ETFs the appropriate Heston's SV model to fit the observed market option data (i.e., on 15 October 2021 option series for each) and derived from it the implied RND of Heston's model (HS). This RND, which is obtained both theoretically, using (A2) and (3), and also via Monte Carlo simulations of (A1), will serve as a benchmark for comparison.

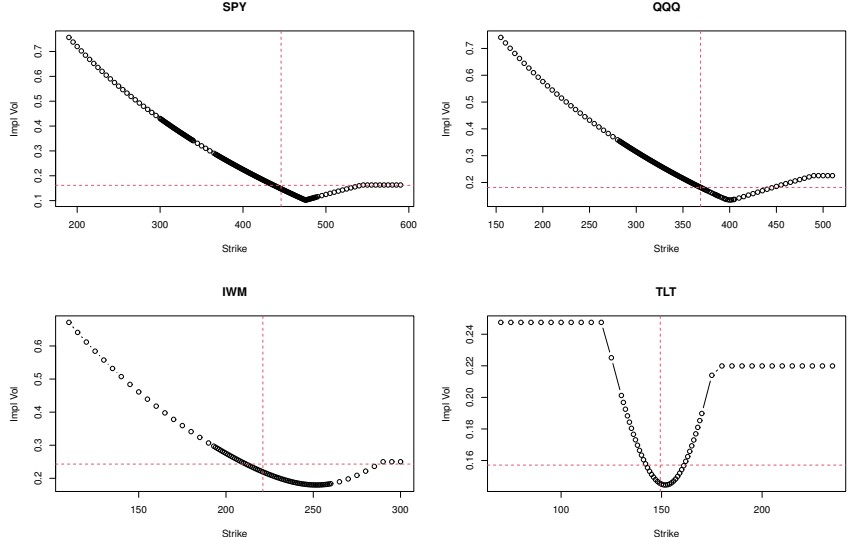

**Figure 1.** The volatility 'smiles' of the the 15 October 2021 option series (calls) as observed and calculated on 13 August 2021 (EOD) for the three market index ETFs, SPY, IWM and QQQ and on 18 August 2021 (EOD) for the TLT ETF.

For each option series, the available market data consist of the $N$ strikes, $K_1, \ldots, K_N$ with corresponding call option (market) prices $C_1, \ldots, C_N$[6]. As a standard measure of the

*goodness-of-fit* between the model-calculated option prices $C^{Model}(K_i)$, $i = 1, \ldots, N$ and the given option market price $C_i$, $i = 1, \ldots, N$, we used the *Mean Squared Error*, MSE,

$$MSE(Model) = \frac{1}{N} \sum_{i=1}^{N} (C^{Model}(K_i) - C_i)^2.$$

Clearly, it is expected that within the scope of the SV model described in (A1), the well-calibrated Heston model will result with a smaller MSE as compared to the Black–Scholes model, so that $MSE(HS) \le MSE(BS)$. However, as we will see below for the available ETF data, pricing the options by a well-calibrated Generalized Gamma (GG) model (19) also resulted with a smaller MSE. In fact, in all four cases, $MSE(GG) \le MSE(BS)$. To demonstrate this, we have taken for each ETF the following steps (conditional of course on the current spot's price $S$ and volatility $V_0$):

- Model Calibration
    - For a given model's parameter, $\vartheta = (\kappa, \theta, \eta, \rho)$ in (A1), we use the `callHestoncf` function of the NMOF package (see Gilli et al. 2019 and Schumann 2011–2021) and the **R** software (R Core Team 2017) to calculate the Heston model's option prices $C_i^{HS}$ for each $K_i$.
    - To calibrate the Heston SV model, we used the `optim(·)` function **R** to minimize $MSE(HS)$ over the model's parameter, $\vartheta = (\kappa, \theta, \eta, \rho)$.
    - For a given $(\alpha, \nu)$ with $\nu = \sigma\sqrt{t}$, we use (19) to calculate the Generalized Gamma model option prices $C_i^{GG}$ for each $K_i$.
    - To calibrate the GG model, we used the `optim(·)` function of **R** to minimize $MSE(GG)$ over the model's parameters, $(\alpha, \nu)$.
    - For a given $\nu$ (where $\nu = \sigma\sqrt{t}$), we use (1) and (2) to calculate the Black–Scholes model option prices $C_i^{BS}$ for each $K_i$.
    - To calibrate the BS model, we used the `optimize(·)` function of **R** to minimize $MSE(BS)$ over the single model's parameter $\nu$ (namely $\sigma$).

- Validation
    - Using the calibrated Heston parameters, $\hat{\vartheta}$, we drew, utilizing a discretized version of Heston's stochastic volatility process (A1), a large number ($M = 30{,}000$) of Monte Carlo simulations, observations on $(S_T, V_T)$ to obtain the simulated rendition of the Heston's RND of $S_t$ (conditional on $S$ and $V_0$, with $t = T - \tau$).
    - Using the calibrated Heston's parameters, $\hat{\vartheta}$, in (A4), we obtain the calculated rendition of the Heston's *theoretical* RND of $S_t$ (conditional on $S$ and $V_0$, with $t = T - \tau$) directly from the characteristics function of $P_2$ (see Appendix A).
    - Finally, we compared all three calibrated risk-neutral distributions of the standardized spot's price (the rescaled spot priced, $S_t^* = S_t/\mu$, where $\mu = Se^{rt}$) as obtained under the Black–Scholes (BS), Generalized Gamma (GG) and Heston (1993) option pricing models (HS).

### 3.2. Calculating the Implied RND under the Volatility Skew

As we mentioned earlier, the data on the 15 October 2021 option series of the SPY, IWM and QQQ were retrieved as of the closing of trading on Friday 13 August 2021 with 63 days to expiration, so that $t = 63/365$ and the prevailing (risk-free) interest rate at that time is $r = 0.0016$. There will be common values for these three highly liquid ETFs. The 15 October 2021 option series of the TLT was retrieved on 18 August 2021 with 57 days to expiration, so that $t = 57/365$ for that ETF. However, we begin our exposition with the details of the largest (volume-wise) of them, namely the SPY. The cases of the IWM, QQQ and TLT will be treated similarly below.

On that day, the closing price of the SPY was $S = 445.92$, and the dividend it pays is at a rate of $\ell = 0.0123$. We incorporate the dividend in our calculations along the lines of Remark 1. The reported (BS-based) implied volatility was $IV = 16.15\%$, which

we will use as our initial value for $V_0$ and for $\sigma$. This option series has $N = 211$ pairs of strike-price $(K_i, C_i)$, which were all used to calibrate Heston's SV model over the model's parameter, $\vartheta = (\kappa, \theta, \eta, \rho)$, with the initial values of $(15, (0.1)^2, 0.1, -0.65)$ and with $V_0 = IV^2 = (0.1615)^2$. The results of the calibrated values are

$$\hat{\vartheta} = (15.03132587, \ 0.02793781, \ 2, \ -0.77469470).$$

This calibrated parameter, $\hat{\vartheta}$, was then used to calculate, using Heston's characteristic function (i.e., (A2)), the option prices according to Heston's SV model (3). This resulted with $MSE(HS) = 0.2226429$. The calibrated (least squares estimate) value of $\sigma$ that minimizes the MSE for the BS model is $\hat{\sigma} = 0.137348$, so that $MSE(BS) = 1.781981$. Accordingly, $\hat{v}_{BS} = 0.137348\sqrt{t} = 0.0570619$ is to be used for the calculation of the $pdf$ of the $\mathcal{N}(-v^2/2, v^2)$ distribution, which leads to the BS formula in (1) (see Example 3.1 in Boukai 2021 for more details). Next, we calibrated the General Gamma distribution according to the pricing model in (19), with initial values of $\alpha = 0.5$ and $\sigma = 0.1615$, which resulted with calibrated value of $\hat{\alpha} = 0.1554312$ and $\hat{\sigma} = 0.1483843$ and an $MSE(GG) = 0.339441$. Clearly, in this case of the SPY, the MSE of the GG pricing model is substantially smaller than the MSE of the Black–Scholes model and is similar to that of Heston's SV pricing model. Indeed, the MSE of the BS model is over 500% as large as those of the GG and the HS models.

To compare the actual distributions, as were calculated under each of these three pricing models, we present in Figure 2 the three implied distributions (as RNDs), which were calculated based on their respective calibrated parameter values. As an added validation, we plotted these three density curves against the histogram of the Monte Carlo simulation of the standardized SPY prices using a discretized version of the pricing model in (A1) (using the calibrated Heston parameters with a seed = 452361). This figure clearly demonstrates the 'inaptness' of the standard BSM Formula (1) and hence the log-normal distribution for the direct (risk-neutral) modeling of option prices in cases which involve negatively skewed price distributions. In fact, the calculated values of the kurtosis and skewness measures of each of these distributions (see Table 1) are also indicative of the noted lack-of-fit of the BS model in these cases and the apparent close agreement of the GG distribution to the exact risk-neutral distribution of the Heston's model and that of the simulated price data.

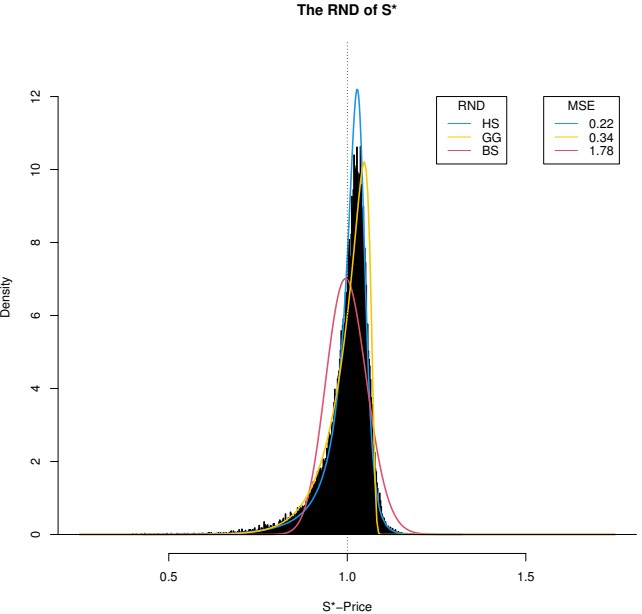

**Figure 2.** The SPY case: the calculated HS, GG and BS implied RNDs along with the Monte Carlo distribution of the spot's price $S^*$ and the corresponding values of the MSEs.

**Table 1.** Calculated (excess) kurtosis and skewness measures for the three distributions depicted in Figure 1 for the SPY option data.

| Measure | HS | GG | BS |
|---------|------|------|------|
| Kurtosis | 7.302674 | 3.536461 | 0.05234164 |
| Skewness | −2.050771 | −1.580122 | 0.1715114 |

The impact of this model's misspecification on the calculated delta values associated with the option series is also of interest. It is a standard practice of the retail brokerage houses to provide, along with the market prices for the option chain, also the BS-base calculated delta for each strike (using some ATM implied volatility value). For example, for the ATM strike of $K = 445$, the quoted delta is $\Delta^* = 0.497$ with a quoted *IV* of 0.1489, whereas under the BS model we calibrated here with $\hat{\sigma} = 0.137348$, we obtained $\Delta_{BS} = 0.506$. However, accounting for stochastic volatility in the pricing model, we calculate for this same strike, $K = 445$, $\Delta_{HS} = 0.663$ by the (better fitting) Heston SV model, and $\Delta_{GG} = 0.638$, by its close proxy, the GG model. Thus, in this case, the BS modeling at the ATM strikes will result with grossly understated delta values (of nearly 25.0%). Without doubt, the impact of this model's misspecification would have profound hedging implications for the retail trader. To fully appreciate the extent of this impact, we present in Figure 3 the values of the delta function (9) as was calculated for the HS model (using $P1$ and (A1)), for the GG model (using (17)) and for the BS model (using $\Phi(d_1)$ from (1)), along with quoted delta values for the SPY chain.

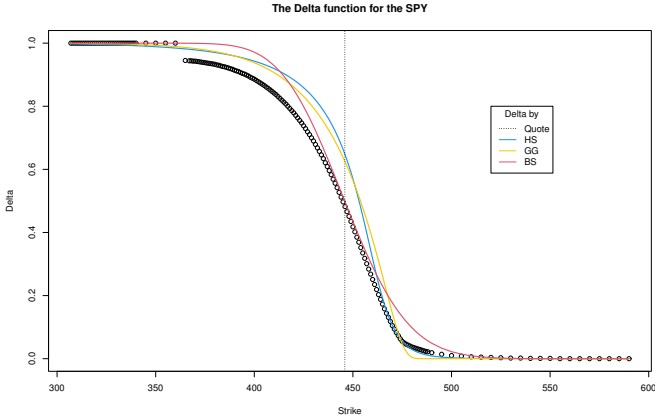

**Figure 3.** The SPY case—the calculated delta functions under each of the pricing models, HS, GG and BS, along with the quoted delta per each strike *K* in the 15 October 2021 option series.

Needless to say, the noted understatement of the quoted (BS -based) delta values as compared to those derived from the SV model also impact the trading strategies. For example, a trader that would sell a 25-delta strangle based on the quoted values will sell the $k_1 = 424$ put for \$5.215 and the $k_2 = 460$ call for \$2.685, collecting a total of \$7.90 for it, which amounts to 21.9% of the spread between the strikes (for a discussion of this ratio, see Boukai 2020). On the other hand, if the trader would have priced the 25-delta strangle according to the GG model (which accounts for the skew), she will sell the $k_1 = 435$ put for \$7.205 and the $k_2 = 466$ call for \$1.435, collecting a total of \$8.64 for it, which amounts to 27.9% of the spread between the strikes, clearly collecting a higher premium for the same 25-delta strangle.

The situation with the other two market index ETFs, IWM and QQQ, is very similar to the one describing the SPY—see the corresponding depiction of their volatility 'smiles' in Figure 1. Following a similar calibration and validation approach, we present (implied) the price distributions derived from the IWM option data shown in Figure 4a and the QQQ option data shown in Figure 5a. The calculated values of the corresponding delta functions are displayed in Figures 4b and 5b. Furthermore, to serve as a contrasting illustration,

we present in Figure 6 the three implied price distributions derived from the TLT ETF option series, along with the corresponding calculated delta functions for that ETF. The situation with the TLT ETF is clearly different, as compared to the three market index ETFs (SPY, IWM and QQQ) which exhibit a pronounced skew of their volatility 'smile'. In the case of the TLT) ETF, with a relatively intact volatility 'smile' (see Figure 1), the implied RNDs are relatively symmetric, and the three option pricing models (HS, GG and BS) yield very similar results. In Table 2, we provide a summary of the *goodness-of-fit* of each of the pricing models as measured by the respective MSE for each of the four ETFs. A corresponding comparison of the ATM delta calculations under each of the option price models is presented in Table 3. In Table 3, we provide a summary of the *goodness-of-fit* as measured by the respective MSE for each of the ETFs. Some of the technical details are provided in Section 4.1 below.

**Table 2.** Model's goodness-of-fit as measured the respective MSE for each of the four ETFs.

| ETF | HS | GG | BS |
|---|---|---|---|
| SPY | 0.2226429 | 0.339441 | 1.781981 |
| IWM | 0.001900968 | 0.01419628 | 0.3750478 |
| QQQ | 0.02418013 | 0.06561134 | 1.193867 |
| TLT | 0.03423748 | 0.04618725 | 0.04341321 |

**Table 3.** Comparison of the the quoted ATM delta $\Delta^*$ of the four market ETF to those calculated under each of the three option pricing models.

| ETF | $S$ | ATM $K$ | $\Delta^*$ | $\Delta_{BS}$ | $\Delta_{GG}$ | $\Delta_{HS}$ |
|---|---|---|---|---|---|---|
| SPY | 445.92 | 445 | 0.497 | 0.506 | 0.638 | 0.663 |
| IWM | 221.13 | 221 | 0.510 | 0.516 | 0.598 | 0.610 |
| QQQ | 368.82 | 369 | 0.507 | 0.503 | 0.625 | 0.632 |
| TLT | 149.35 | 150 | 0.511 | 0.453 | 0.477 | 0.467 |

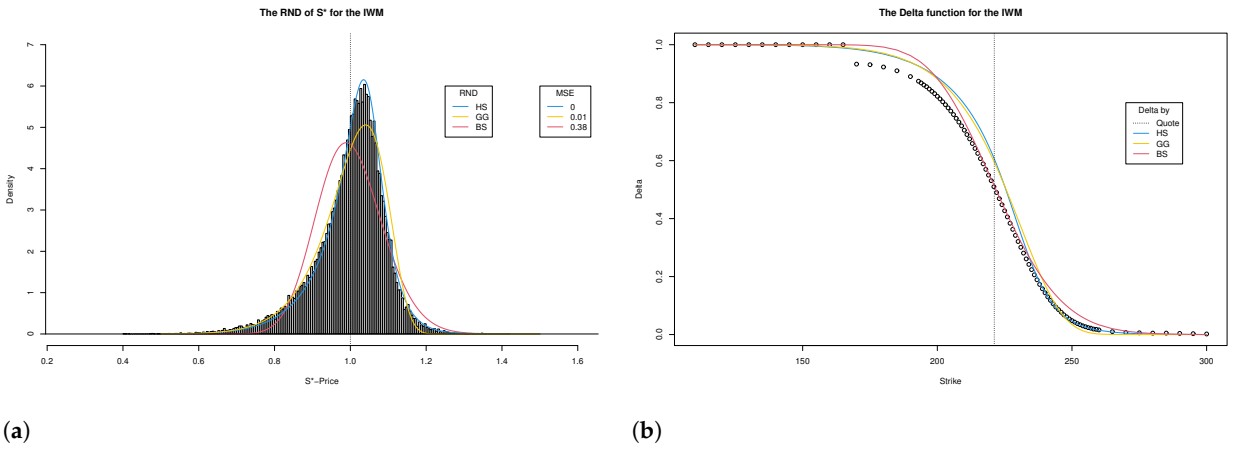

(**a**)           (**b**)

**Figure 4.** The IWM case: (**a**) the HS, GG and BS implied RNDs along with the Monte Carlo distribution of the Spot's price $S^*$, and (**b**) the corresponding delta functions along with the quoted delta per each strike $K$ in the option series.

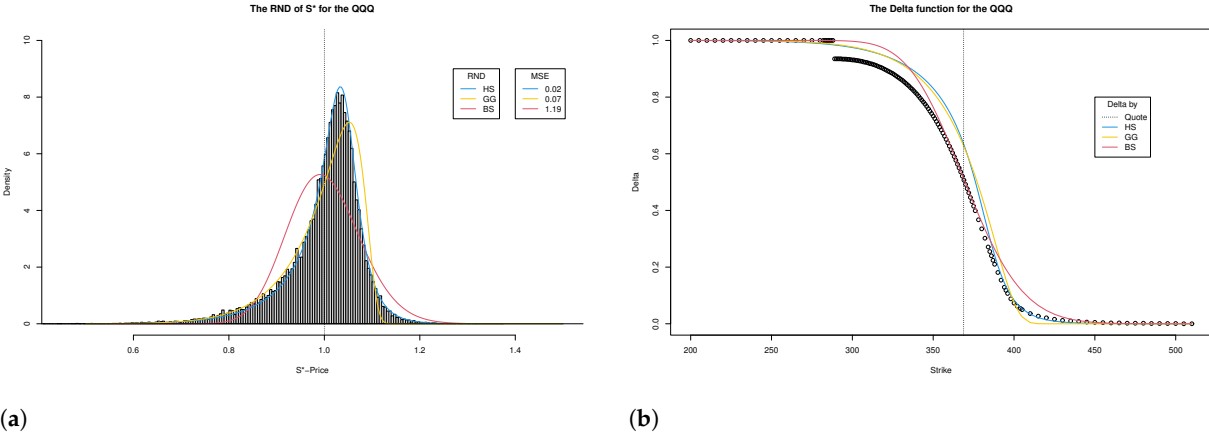

(**a**)　　　　　　　　　　　　　　　　　　　(**b**)

**Figure 5.** The QQQ case: (**a**) the HS, GG and BS implied RNDs along with the Monte Carlo distribution of the Spot's price $S^*$, and (**b**) the corresponding delta functions along with the quoted delta per each strike $K$ in the option series.

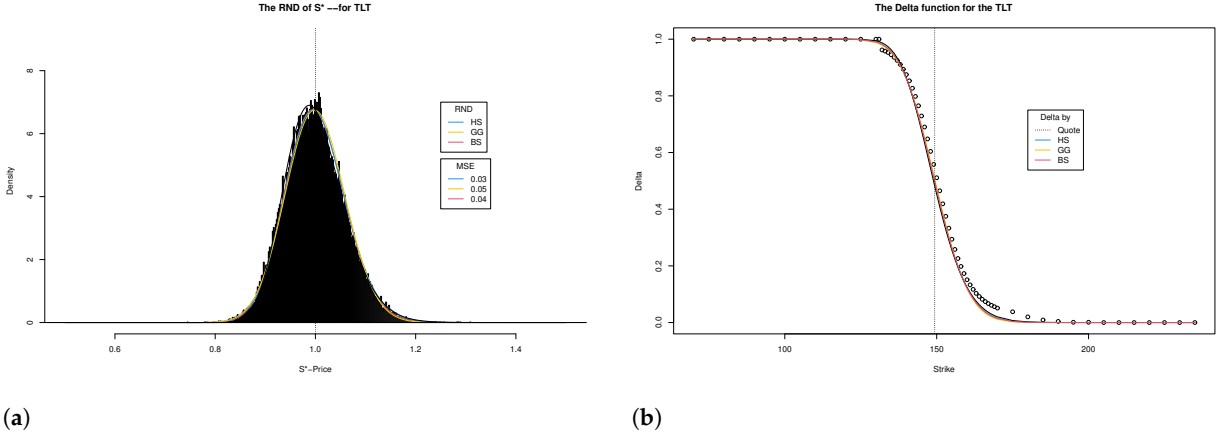

(**a**)　　　　　　　　　　　　　　　　　　　(**b**)

**Figure 6.** The TLT case- (**a**) the HS, GG and BS implied RNDs along with the Monte-Carlo distribution of the Spot's price $S^*$, and (**b**) the corresponding delta functions along with the quoted delta per each strike $K$ in the option series.

## 4. Summary and Discussion

As was illustrated in all the above examples, the Heston (1993) option pricing model (as given in (A1) and (3)), which accounts for the presences of stochastic volatility, produces as expected the best results overall as compared to the Black–Scholes option pricing model (1) with its presumed constant volatility. Clearly, a well-calibrated Heston model will always result in a better fit to realistic market option data (indeed, resulting with $MSE(HS) < MSE(BS)$) and would be the default modeling choice for the practitioner. Unfortunately, however, the numerical challenges involved in the calculations and calibration (or optimization of $\vartheta = (\kappa, \theta, \eta, \rho)$) process of the Heston's option pricing model (see for example Romo and Ortiz-Gracia 2021 or Lemaire et al. 2020) render it largely inaccessible to many of the retail option traders who do not possess the prerequisite skills or know-how to meet these numerical challenges. In comparison, the calculations and calibration process involved with the two-parameter Generalized Gamma distribution as RND for Heston's SV option valuation are substantially simpler and more straightforward (and could potentially be accomplished within an Excel spreadsheet). As was demonstrated earlier, the GG model is significantly more accurate than the Black–Scholes model for the pricing of the options in a skewed stochastic volatility environments as those exhibited (at present times) by the three markets ETFs, SPY, IWM and QQQ. In fact, in situations that imply negatively skewed price-distributions as RND, the Black–Scholes pricing model, and hence the log-normal distribution as RND, will surely be inferior to the GG distribution as an RND and surely

to Heston's SV pricing model in fitting realistic option market data. In such situations, one would realize (as we did in these examples) $MSE(GG) < MSE(BS)$ and would want to adopt the GG RND for the underlying pricing model. In contrast, in situations such as the one exhibited by the `TLT` ETF, one would realize $MSE(GG) \approx MSE(BS)$, as all three option pricing models (including Heston's) produce similar results. Although not expressly covered by the examples we included here, we have grounds to believe that the same conclusion could be arrived upon using the Inverse Generalized Gamma (see Section 2.2) in situations involving positively skewed (implied) RND in the option pricing model. Although the two-parameter versions of the GG and IGG distribution are similar, they differ in their implied parameter space (see restrictions in (13) and Section 2.3) and thus should be treated separately. In all, both of these versions of the Generalized Gamma distribution could serve as useful proxies to the exact Heston's RND given in (A4) and hence produce superior results to those obtained by the Black–Scholes model in an environment involving stochastic volatility. Thus, given the market option data, one could simply calculate $MSE(BS), MSE(GG)$, and if appropriate also $MSE(IGG)$, and adopt the RND for the option pricing model, which produces the smaller $MSE$ and hence the better fit to the market data.

### 4.1. Some Technical Notes

- The 15 October 2021 option series data files `SPY_63.csv`, `IWM_63.csv`, and `QQQ_63.csv` as were obtained on the EOD of 13 August 2021 and that of `TLT_57.csv` obtained at the EOD of 18 August 2021 are available from the author upon request. Their basic summary information is provided in Table 4 below.

**Table 4.** Summary information of the four ETFs.

| ETF | $S$ | DTE | $N$ | Quoted IV | Div. Rate |
|-----|-----|-----|-----|-----------|-----------|
| SPY | 445.92 | 63 | 211 | 16.15% | 1.23% |
| IWM | 221.13 | 63 | 93 | 24.30% | 0.63% |
| QQQ | 368.82 | 63 | 160 | 18.13% | 0.43% |
| TLT | 149.35 | 57 | 66 | 15.71% | 1.46% |

- The standard **R** function `dgamma` and `pgamma` were used to calculate the *pdf* and *cdf* in (11) and hence used in the calculation of (19); see Appendix B.
- The `cfHeston` and `callHestoncf` functions of the NMOF package of **R** were used in the calculation of (A2) and (3).
- A modification of the `callHestoncf` function of the NMOF package of **R** was used to calculate (A4).
- The `optim` and `optimize` functions of **R** were used in the calibration of the three models (HS, GG and BS) for the available option data.
- The initial and the calibrated values of $\vartheta = (\kappa, \theta, \eta, \rho)$ of the Heston's model were:

  SPY: $(15, (0.1)^2, 0.1, -0.65)$ and $(15.03132587, 0.02793781, 2, -0.77469470)$.
  IWM: $(5, (0.1)^2, 0.6, 0)$ and $(4.97834286, 0.04032166, 1.09837930, -0.59905916)$.
  QQQ: $(3.5, (0.2)^2, 0.5, -0.5)$ and $(3.47635183, 0.06382197, 1.13505528, -0.69137767)$.
  TLT: $(3, (0.1)^2, 0.1, 0.1)$ and $(2.99997881, 0.01459405, 0.10011507, 0.10007980)$.

- For the Monte Carlo simulation of (A1), we employed the (reflective version of) Mil'shtein (1975) discretization scheme (see also Gatheral 2006) with seeds = 4569 (QQQ), = 777999 (IWM), = 452361 (SPY) and = 121290 (TLT).

**Funding:** This research received no external funding.

**Institutional Review Board Statement:** Not applicable.

**Informed Consent Statement:** Not applicable.

**Data Availability Statement:** The data are available upon request from the author.

**Conflicts of Interest:** The author declares no conflict of interest.

## Appendix A. Heston's 1993 Solution

Heston (1993) considered the stochastic volatility model describing the price-volatility dynamics (of $\mathcal{S} = \{S_t, t \geq 0\}$ and $\mathcal{V} = \{V_t, t \geq 0\}$) as described via a system of stochastic deferential equations (SDE) given by

$$
\begin{aligned}
dS_t &= rS_t dt + \sqrt{V_t} S_t dW_{1,t} \\
dV_t &= \kappa(\theta - V_t) + \eta\sqrt{V_t} dW_{2,t},
\end{aligned}
\tag{A1}
$$

where $r$ is the risk-free interest rate, $\kappa$, $\theta$ and $\eta$ are some constants (as discussed in Section 1) and where $W_1 = \{W_{1,t}, t \geq 0\}$ and $W_2 = \{W_{2,t}, t \geq 0\}$ are two Brownian motion processes (under the risk neutral probability $\mathbb{Q}$) with $d(W_1 W_2) = \rho dt$ for some $\rho^2 \in (0, 1)$). Heston (1993) offered $C_S(K)$ in (3) as the solution to the option valuation under the above SDE and provided (semi) closed form expressions to the probabilities $P_1$ and $P_2$ that comprise it. These closed form expressions are given for $j = 1, 2$ by

$$
P_j = \frac{1}{2} + \frac{1}{\pi} \int_0^\infty \mathcal{R}e\left[\frac{e^{-i\omega k}\psi_j(\omega, t, v, x)}{i\omega}\right] d\omega,
\tag{A2}
$$

where with $x := \log(S)$, $k := \log(K)$, $b_1 = \kappa - \rho\eta$, $b_2 = \kappa$ and $\psi_j(\cdot)$ is the characteristics function

$$
\psi_j(\omega, t, v, x) := \int_{-\infty}^\infty e^{i\omega s} p_j(s) ds \equiv e^{B_j(\omega,t) + D_j(\omega,t)v + i\omega x + i\omega rt}.
$$

Here, $p_j(\cdot)$ is the *pdf* of $s_T = \log(S_T)$ corresponding to the probability $P_j$, $j = 1, 2$ and

$$
B_j(\omega, t) = \frac{\kappa\theta}{\eta^2}\left\{(b_j + d_j - i\omega\rho\eta)t - 2\log(\frac{1 - g_j e^{d_j t}}{1 - g_j})\right\}
$$

$$
D_j(\omega, t) = \frac{b_j + d_j - i\omega\rho\eta}{\eta^2}\left(\frac{1 - e^{d_j t}}{1 - g_j e^{d_j t}}\right)
$$

$$
g_j = \frac{b_j - i\omega\rho\eta + d_j}{b_j - i\omega\rho\eta - d_j}
$$

$$
d_j = \sqrt{(i\omega\rho\eta - b_j)^2 - \eta^2(2i\omega u_j - \omega^2)}.
$$

Now, by a standard application of the Fourier transform, we obtain (see for example Schmelzle 2010) that the *pdf* $p_j(\cdot)$ of $s_T = \log(S_T)$ can be computed, for any $s \in \mathbb{R}$, as

$$
p_j(s) = \frac{1}{\pi} \int_0^\infty \mathcal{R}e\left[e^{-i\omega s}\psi_j(\omega, t, v, x)\right] d\omega.
\tag{A3}
$$

Hence, it follows immediately that the *pdf* $\tilde{p}_j(\cdot)$ of $S_T$ is given, for any $u > 0$, by

$$
\tilde{p}_j(u) = \frac{1}{u} \times p_j(\log(u)) \equiv \frac{1}{\pi} \int_0^\infty \mathcal{R}e\left[\frac{e^{-i\omega \log(u)}\psi_j(\omega, t, v, x)}{u}\right] d\omega.
$$

Further, since the *c.f.*, $\psi_j$, above are affine in $x + rt = \log(S) + rt \equiv \log(\mu)$, we may rewrite $\tilde{p}_j(u)$ as

$$
\tilde{p}_j(u) = \frac{1}{\mu\pi} \int_0^\infty \mathcal{R}e\left[\frac{e^{-i\omega \log(u/\mu)}\tilde{\psi}_j(\omega, t, v)}{u/\mu}\right] d\omega,
\tag{A4}
$$

where $\log(\tilde{\psi}_j(\omega, t, v)) := \log(\psi_j(\omega, t, v, x) - i\omega x - i\omega rt$. Note that $\tilde{p}_2(\cdot)$ in (A4) is the *p.d.f* corresponding to the RND, $Q_\mu(\cdot)$, of $S_T$ (under $\mathbb{Q}$) for Heston's (1993) model with

$P_2 \equiv \int_K^\infty \tilde{p}_2(u)du = \mathbb{Q}(S_T > K) \equiv Q_\mu(K)$, in (3) and (7). In addition, note that this $Q_\mu(\cdot)$, constitutes a scale-family of distributions in $\mu = Se^{rt}$, so that it satisfies the terms of Assumption 1. As was mentioned in Section 4.1, the cfHeston and callHestoncf functions of the NMOF package of **R** are readily available to accurately compute the values of $\psi_j$ and hence of $P_j$ (as well as $\tilde{p}_j(u)$) as well as the call option values $C_S(K)$ in (3) for given $t, s$ and $v$ and any given choice of $\vartheta = (\kappa, \theta, \eta, \rho)$.

**Appendix B. R Code for the GG Model**

The following is the simple **R** code for calculating the option call price under the GG model as given in (19).

```
##
# s0    #= current spot's price
# k     #= strike
# te    #= days/365
# r     #= interest rate
# q     #= dividend rate
# sig   #= volatility (sigma)
# alpha #= first shape parameter, alpha
##
GG.value<-function(s0, k, te, r, q, sig, alpha) {
    nu<-sig*sqrt(te)
    k<-k*exp(-r*te)
    s0<-s0*exp(-q*te)
    s1<-k/s0
    f0<-GG.call(s1, nu, alpha)
    return(f0)
}
###
GG.call<-function(s, v, alpha){
    xi<-seq(0.1, 100, length=10000)
    yy<- (gamma(alpha)*gamma(alpha+2/xi))/(gamma(alpha+1/xi))^2
    xi0<-min(xi[yy<1+v^2])       # second shape parameter
    lam0<-gamma(alpha)/gamma(a)   # the scale parameter
      s1<-(s/lam0)^(xi0)
      delta<-1-pgamma(s1, alpha+1/xi0, 1)
      prob<-1-pgamma(s1, alpha,1)
      cc0<-delta-s*prob
      f0<-cbind(s, cc0, delta, prob, xi0, lam0)
    return(f0)
}
##
```

**Notes**

[1]  Nowadays, many of the retail brokerage houses operate entirely within the 'Black–Scholes world' and provide, aside from market option bid–ask prices, the 'theoretical price' and related 'Greeks', and implied volatility values as are derived from and calculated under the BSM Formula (1) and (2).

[2]  Note that $c_\mu(\cdot)$ is merely the *undiscounted* version of $C_S(\cdot)$ in (4). For the linear homogeneity property of the European options, in general, see for example Theorems 6 & 9 of Merton (1973).

[3]  As of the original draft of this paper, 14 August 2021.

[4]  As of the writing of this paper, 14 August 2021.

[5]  Option chain quotes were retrieved from TD Ameritrade using the TOS platform.

[6]  These prices could be the actual market prices or the average between the bid and ask prices of the market.

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
