# Peer review of "The Generalized Gamma Distribution as a Useful RND under Heston’s Stochastic Volatility Model"

_jrfm, doi:10.3390/jrfm15060238_

Round 1

Reviewer 1 Report

It was a pleasure to read this very well-written article.

One important addition is that the author should cite the NMOF package, using both the book and package, as requested by the developers (https://cran.r-project.org/web/packages/NMOF/citation.html).

It could be argued that the inclusion of the IGG is unnecessary in this article. (Remove section 2.2 and other associated references to it.) However, the author notes that it is included for completeness, and I suppose it doesn't really matter one way or the other.

I regret that my review adds little value to this process, but the article is truly in excellent shape already and I recommend publication.

Reviewer 2 Report

Summary

The paper studies the generalized Gamma distribution in a Heston model. The author argues that this approach as benefits in comparison to the Black-Scholes(-Merton) model under a GBM assumption and the Heston model.

The topic is of interest to the journal, yet the paper needs to be clarified on the methodological part.

Major Comments:

  • The Black-Scholes model is well-known in the literature and there is no need to discuss equ. (1) and (2) such prominent given you focus on the Heston model and Gamma distribution. I recommend to start with your (new) study object in line 29!
  • The literature is not up-to-date. I think the author have to do a thorough literature research in that field, particularly research work over the past years: 2012 to 2020, 2021, 2022.
  • The methodological argument hinges mainly on Boukai (2021) in this paper. Yet, this is a non-peer-reviewed SSRN paper. In this case, the author must convince the reader/referee that this approach is reasonable. Please show all arguments/proof in the main text or appendix. Show – as you try to do on page 4ff. – yet not in a rigorous fashion so far. Indeed, show equ. (8)ff. à So far, this is the weakest point in the paper. Moreover, Q(K) is not sufficiently defined in equ. (6) – later it becomes clearer.
  • On p. 6ff. I would be more explicit for the reader. Include short proofs in the appendix for the moments, such as , etc. Sorry, ‘it can be easily verified’ is your job before publication in order to convince the reader/referee (equs. 12-17, p. 6). To my reading, the inverse gamma distribution in section (2.2) is rather a re-labelling if the above part is better and rigorously developed.

In my view, the idea and model are intriguing and interesting. Yet, the author has to develop the mathematical concepts until page 6 more systematic (step-by-step). Currently the whole section is a bit confusing and jumps - without step-be-step derivation or proofs.

  • In order to strengthen the empirical part, I recommend more data points and analysis; not just August: Give you pretend the GG approach is easily doable in comparison to HS. Moreover, the HS model is still good and probably better than the GG approach (Figure 2 etc.) or rather similar. Sorry, the BS model is not a real honest benchmark in adv. theory and practice. Your benchmark is the HS-model. Of course, you discuss some issues in lines 330ff., yet a new model should improve something. The argument is easier not so convincing, given that the methodological part is still a bit confusion. And: Why all the labour if HS model still does the better?

Note: good science requires transparency. Thus, you have to reveal all data and code. Please upload the data and R-code or copy it to the appendix.

Minor Comments

  • Abstract does not follow scientific conventions: (i) major field, (ii) major research question different to the literature until 2022, (iii) findings and difference to literature and (iv) conclusions. All has maximum 150 to 200 words!
  • Follows in several cases in your paper: please introduce shorthand’s always with the full name at the first occurrence: First example: PDE? = partial differential equation (PDE) in line above equ. (3).
  • Wording: ‘complex integration’ – do you really do integration of holomorphic functions over a complex number? I think you work with the characteristic function that contains a complex number – but nothing else. Please, avoid the word of ‘complexity’ because it is wrongly used. You neither have a ‘complex system’ nor something is ‘complex’ in your work. Everything should probably from a mathematical point of view obvious and basic at an adv. level.
  • Subsection 1.2 starts with repetitions. Please make a thorough text editing.
  • Typo in line 293.

Round 2

Reviewer 2 Report

Dear Author(s):

Finally, I recommend to proofread the paper and do minor text editing. 

Regarding BS model: I still think this part is replication and not required. At least an expert is bored by the beginning. 

Uploading the non-peer-reviewed paper Boukai (2021) as a technical appendix including the databox is highly recommended. Indeed, it is needed to provide a thorough understanding for this approach. Or integrate the proofs/paper by Boukai (2021) fully in the appendix of this paper.
